# Recent Advances of Microbiome-Associated Metabolomics Profiling in Liver Disease: Principles, Mechanisms, and Applications

**DOI:** 10.3390/ijms22031160

**Published:** 2021-01-25

**Authors:** Ganesan Raja, Haripriya Gupta, Yoseph Asmelash Gebru, Gi Soo Youn, Ye Rin Choi, Hyeong Seop Kim, Sang Jun Yoon, Dong Joon Kim, Tae-Jin Kim, Ki Tae Suk

**Affiliations:** 1Institute for Liver and Digestive Diseases, Hallym University, Chuncheon 24252, Korea; RG@hallym.ac.kr (G.R.); phr.haripriya13@gmail.com (H.G.); yagebru@gmail.com (Y.A.G.); gisu0428@hallym.ac.kr (G.S.Y.); dpfls3020@gmail.com (Y.R.C.); kimhs2425@gmail.com (H.S.K.); ysjtlhuman@gmail.com (S.J.Y.); djkim@hallym.ac.kr (D.J.K.); 2Department of Biological Sciences, Integrated Biological Science, and Institute of Systems Biology, Pusan National University, Busan 46241, Korea; tjkim77@pusan.ac.kr

**Keywords:** metabolomics, metabolic engineering, discriminations, liver therapies, gut microbiome, scientific applications

## Abstract

Advances in high-throughput screening of metabolic stability in liver and gut microbiota are able to identify and quantify small-molecule metabolites (metabolome) in different cellular microenvironments that are closest to their phenotypes. Metagenomics and metabolomics are largely recognized to be the “-omics” disciplines for clinical therapeutic screening. Here, metabolomics activity screening in liver disease (LD) and gut microbiomes has significantly delivered the integration of metabolomics data (i.e., a set of endogenous metabolites) with metabolic pathways in cellular environments that can be tested for biological functions (i.e., phenotypes). A growing literature in LD and gut microbiomes reports the use of metabolites as therapeutic targets or biomarkers. Although growing evidence connects liver fibrosis, cirrhosis, and hepatocellular carcinoma, the genetic and metabolic factors are still mainly unknown. Herein, we reviewed proof-of-concept mechanisms for metabolomics-based LD and gut microbiotas’ role from several studies (nuclear magnetic resonance, gas/lipid chromatography, spectroscopy coupled with mass spectrometry, and capillary electrophoresis). A deeper understanding of these axes is a prerequisite for optimizing therapeutic strategies to improve liver health.

## 1. Introduction

In the nineteenth century, the term “metabolomics” was used for the first time and was defined as the quantitative and qualitative analysis of small molecules/metabolites (low molecular weight molecules, <1500 Da) and their patterns in cells. Among the ‘-omics’ sciences, metabolomics (also stated as metabolomics profiling and metabonomics) has been so technologically advanced that it acts as a division of systems biology [1,2]. The high-throughput global analysis of metabolomics profiling has placed it at final step in the -omics cascade. The metabolic profiling technologies can focus on metabolic stability, metabolites structure, target profiling, and associated metabolic pathways. Metabolomics profiling and chemical profiling are able to investigate the mechanical properties of metabolome (full set of metabolites within cells or tissue) in molecular networks and have been routinely applied as tools for clinical therapeutics [3,4].

Metabolomics methods involve a comprehensive analysis of small-molecule metabolites under a given set of conditions. Metabolites serve as direct signatures of metabolic reaction and biochemical activity. The metabolome is the full set of metabolites within a given cell type of tissue or cells. The metabolite concentration is directly connected with phenotypic expression, which acts as a functional endpoint of metabolisms and reflects genetic activity (gene expression) and protein activity (proteome) [5,6]. Representing every chemical reaction, metabolites play the important role of running the metabolic pathways. The applications of metabolomics have continually grown, which has led to refinement of methods for measurement, analysis, and understanding of complex data sets [2]. The specific metabolic pathways can be discovered and can underlie key therapies for liver diseases, gut malfunctions, and avoidance of liver burden.

Metabolomics is a collection of measurements performed on biological samples for quantifying the several metabolites (metabolome) and evaluating fluctuations in metabolite levels [7]. Every single cell is transmitted over four different stages—genomics, transcriptomics, proteomics, and metabolomics (Figure 1). Among them, metabolomics has a particular advantage over other omics technologies. Metabolic target profiling and global metabolic profiling (untargeted metabolomics) have delivered molecular phenotypic variability that acts a therapeutic agent for specific diseases [7,8,9]. However, both targeted and untargeted metabolomics techniques in the metabolite pool play very important roles in heterogeneous cancer evolution. Moreover, both targeted and untargeted profiling can provide a better idea of the cellular conditions and molecular messages for the representation of cellular phenotypes. Additionally, they show wonderful applications in biomarker discovery [10,11].

Appropriate analytical tools that allow comprehensive and robust metabolic analysis of various chemical structures is key to understanding the impact of metabolic signatures. Nuclear magnetic resonance (NMR), gas chromatography (GC), liquid chromatography (LC), Fourier transform infrared spectroscopy (FTIR) coupled with mass spectrometry (MS: LC/GC-MS), and capillary electrophoresis (CE-MS) have frequently existed in metabolomics profiling for metabolic stability and structural characterization of metabolome from cells, tissues, plant extracts, gut microbiota, and bacterial samples [12,13]. Analytical applications of NMR, LC/GC-MS, CE-MS, and FTIR were compared to investigate and assess the quality of analysis of given biofluids (Table 1). The dynamic changes in metabolite profile behind NMR spectroscopy and LC/GC-MS analytical technologies are used to generate a spectral profile according to intensities and mass-to-charge (*m*/*z*) ratios, respectively, from which can be extracted information that pertains to physiology and latent disease. Each technology has individual applications. However, leading machines of NMR and LC/GC-MS methods provide good quality of metabolomics large datasets [12,14].

The gut microbial community consists of an eco-system with functions in host protection against pathogens, immune modulation, metabolic pathways, and enterohepatic circulation. As the liver is directly connected with the gut through the portal vein, gut-derived toxic factors including bacteria, damaged metabolites (damage-associated molecular patterns), or bacterial products (pathogen-associated molecular patterns) are metabolized in the liver [15]. Some gut microbiota produce ammonia, ethanol, and acetaldehyde, which are mostly metabolized in the liver and are associated with Kupffer cell activation and the inflammatory cytokine pathway [16]. Interest in metabolites is increasing as gut microbiome-related metabolites are a key pathophysiologic factor in liver disease (LD) progression.

Gut bacteria genera might be involved in the fermentation biology of polysaccharides, energy collection, bile acid (BA) synthesis, and choline metabolism in liver cells. Bacterial changes such as *Enterobacteria, Bacteroides, Proteobactteria, Faecalibacterium, Ruminococcus, Lactobacillus,* and *Bifidobacterium* have been mirrored in liver metabolites and their metabolic reaction network. Liver cirrhosis is related to bile secretion disorders and metabolic syndrome. Gut microbiota have been involved in the amelioration of liver diseases. In this review, the analysis technology, specific metabolites, and application fields of metabolomics in liver disease are described.

## 2. Defect Metabolomics and Molecular Phenomics of Liver Diseases

### 2.1. Overview

Metabolomic profiling and molecular analysis have attempted to characterize the transformation of the liver from a healthy to a diseased state (i.e., fatty liver diseases (FLD), non-alcoholic steatohepatitis (NASH), fibrosis, hepatic steatosis, and cirrhosis). In the liver cellular environment, metabolic transformations take place that may activate or deactivate the molecules. In most cases, liver metabolic changes produce several metabolites that will directly be usable by other organs. Figure 2 shows the frequency of articles published on metabolomics and liver diseases, respectively. This review covers only literature published between 2010 and 2020 (21 September 2020). A schematic diagram of metabolomics profiling (NMR, LC/GC-MS) and metabolic engineering pipeline is displayed in Figure 3.

At the initial stage, over the 5% of fat deposition in liver is called fatty LD and is divided into two phases: alcoholic fatty LD (AFLD) and non-AFLD (NAFLD) [28]. Until now, FLD has not been a serious issue, although FLD is fundamental for NASH, cirrhosis, hepatocellular carcinoma [29,30]. LD covers a large field of diseases, from asymptomatic FLD to NASH and cirrhosis. Basically, fatty liver is formed when more high-fat food, alcohol, and high sugar diets are consumed. Here, every food has a different pathogenic cellular metabolism. In this review, we will focus on the major stages of FLD [31,32].

### 2.2. Alcoholic Fatty Liver Diseases and Metabolome Phenomics

There are three mainstream high-data content metabolic profiling platforms. Programmed high-throughput metabolomics platforms are used to identify the concentrations of metabolites, organic compounds, their associated spectral data, and their metabolic flexibility. From the recent updates, alcohol-induced fatty liver diseases and their molecular mechanisms are complex and could not be fully studied. According to alcohol dehydrogenase (ADH) level, alcohol metabolites are degraded and form acetaldehyde, which may be toxic to cells [33]. After this, it is metabolized to acetate by aldehyde dehydrogenase. The reduced level of ADH may influence alcohol consumption [34]. For bioactive fatty acid and cholesterol biosynthesis, acetate, acetyl-CoA and acetoacetyl-CoA can be involved, which may lead the fatty liver expansion [35].

A liquor may promote free radicals (i.e., superoxide, O_2_^−^; hydroxyl radicals, OH; nitric oxide, NHO; organic radicals, R^•^; peroxyl radicals, ROO; alkoxy radical, RO; and nitrogen dioxide, NO_2_) and non-radicals (i.e., hydrogen peroxide, H_2_O_2_; singlet oxide, O_2_; ozone, O_3_; and peroxynitrile, ONOH^−^) in liver cellular environments. The growth of reactive oxygen species (ROS) and reactive nitrogen species (RNS) generates oxidative stress. Those radicals of ROS and RNS may affect mitochondrial functions and reduce mitochondrial fat oxidation metabolism, which could give rise to obese fat gathering in liver [11,36]. The ROS and RNS have a major role in oxidative stress/damage to metabolic stability and biological disturbances that boost lipid toxicity and pro-inflammatory cytokines, pro-inflammatory tumor necrosis factor-a (TNF-a), interleukin (IL)-1, IL-1 beta, IL-6, and IL-8. The characterization and the expression of CXCR1 and CXCR2 receptors of cells act as leading candidate biomarkers in liver inflammations [37,38].

In lipid metabolism, sterol regulatory element-binding protein-1 (SREBP1) and fatty acid synthase (FAS) act as leading regulating factors. The inhibition of SREBP-1 and FAS has been shown to prevent AFLD [39,40]. The activation of peroxisomes proliferation-activated receptor-a (PPAR-a) is done by alcohol exposure and quickens synthesis of fatty acids, resulting in AFLD [41]. Finally, alcohol intake leads to the dysbiosis of the gut and increases intestinal permeability, which may promote lipopolysaccharides (LPS) to the liver. Kupffer cells activation plays an important role in liver inflammation [42]. Alcohol-treated urine and liver metabolites are targeted and quantified by 1H-NMR-based metabolomics examination (Figure 4).

### 2.3. Metabolic Phenome of Non-Alcoholic Fatty Liver Disease

Worldwide, NAFLD is the most common chronic LD and is increasingly found around the world, especially in western nations. The advanced stages of chronic LD deliver hepatic inflammation and fibrosis that is named as NASH. NASH has led to cirrhosis, liver failure, and liver cancer [44,45]. With the help of a major inflammatory component, NASH acts in advanced stages of NAFLD. NAFLD arises when fat deposition builds up in the liver [46]. In the general population, NAFLD may lead to the growth of NASH. As a result, 80% of cases continue as isolated fatty liver with no slight development to advanced scarring (cirrhosis). Over 11% of NASH cases developed cirrhosis within 15 years. Similarly, 7% of hepatocellular carcinoma (HCC) was found after six years, whether via direct mechanisms or cirrhosis [47].

The hepatic inflammation of NASH plays a main role in visceral adipose, which another theory noticed that hepatitis also initiates [46]. High-fat diet applied mice studies have supported this theory [48]. Metabolomics analysis addressing metabolic profiling of NASH and NAFLD in serum/plasma samples has been examined. In NAFLD, several fatty acids and triacylglycerols were upregulated in plasma samples [49,50]. In serum, three phospholipids were significantly altered when a sample of NASH was compared with NAFLD [51].

The advanced metabolomic profiling and chemical profiling of NASH pathogenesis has come from recent investigations that jointly analyzed analytical and molecular targeted gene expression [52]. In current learning, NASH may develop in mice when treated with MCD (methionine and choline deficient diet). Metabolomic profiling analysis by ultra-performance liquid chromatography (UPLC; a combination of a 1.7 μm reverse-phase packing material and a chromatographic system)-ESI-TOFMS has been significantly resulted in a reduction of lysophosphatidylcholine (LPC: 16:0), LPC (18:0), and LPC (18:1) in serum, with significant growth in tauro-β-muricholate, taurocholate, and 12-HETE for MCD fed mice, compared to control diet.

Serum of both galactose-amine (GalN)- and saline-applied ob/ob steatotic mice with an MCD diet has shown the same variations in LPC and BAs. In healthy data, genetic activity modified ob/ob mice with GalN-exposure served as steatosis, which improved severe inflammation and hepatocyte injury. The hepatic mRNA’s coding for TNF-α and TNFβ1 increased their levels. The basic quantification of predefined major metabolome and organic acids in liver metabolisms and gut microbial metabolites are shown in Table 2.

So far, in serum, BAs and LPC play a role as the best candidate biomarkers for the inflammatory component of NASH, rather than the steatosis section. Collectively, these molecules explain how serum/plasma metabolites are involved in the inflammatory phenotype of NASH in mouse modal outcomes. Related, similar changes have been found in NASH patients, signifying that comparable molecular mechanisms might happen in humans [49]. Lastly, clinical therapeutic biomarkers for NASH are limited, which may be essential in the metabolomics profiling highlighted in this sector.

### 2.4. Signatures of Liver Fibrosis, Cirrhosis, and Metabolic Phenomics

The summary of metabolomic profiling systems (e.g., NMR, LC/GC-MS, CE-MS, HPLC) from various biofluids and clinical applications of liver fibrosis and cirrhosis is summarized in Table 3. Liver fibrosis acts as a basic stage of liver scarring. It happens by the unnecessary deposition of extracellular matrix proteins, including collagen that arises in the many types of chronic liver diseases (e.g., cirrhosis). Oxidative stress provokes the inflammatory responses and apoptosis involved in cirrhosis [53]. It is very likely that NAFLD/NASH may be responsible for cirrhosis development, because the pathophysiological discovery of NASH in the cirrhotic liver has been inspiring to diagnosis and therapeutic screening [54].

Liver extracts were analyzed by NMR and discussed the enhancement of lactate level, [53] which may deliver an anaerobic metabolism in the fibrotic liver cellular microenvironment. Metabolic phenotype by 1H-NMR spectra at 600 MHz for the skeletal muscle, liver, and serum samples has been characterized from germ-free, pathogen-free, and conventionalized mice [55]. Hepatotoxins provoked fibrosis and cirrhosis and has been confirmed with three different studies in rats. Examining tissues and cellular lipid accumulations under histopathology has demonstrated that thioacetamide in drinking water has settled hepatic fibrosis and cirrhosis in rats.

Carbon tetrachloride (CCl4) induced fibrosis in treated rats [56,57], and the scientists examined protection by the Chinese medicine xia-yu-xue decoction [56] or scoparone, a drug isolated from medicinal plants [57]. After CCl4 exposure, several metabolite signals were labelled and urinary excretion of certain amino acids and gut flora metabolites (which were mostly reversed by xia-yu-xue decoction) were meaningly reduced [56]. Also, the urinary excretion of glycocholate was increased [57]. Here, hepatic fibrosis provoked in a healthy, rather than fatty rat liver, was linked with slight variations in the urinary metabolome.

In total, an eight metabolomics profiles and pattern recognition of hepatic cirrhosis was examined on human biofluids: one on feces [64], one on liver biopsies [58], and six on serum [61,65]. From those examinations, no clear images were found. From serum samples, amplified nonessential amino acids [60], certain D-amino acids [62] and lowered essential amino acids [61,62] strongly suggested that the cirrhotic liver metabolic process had weakened the metabolic capability of both D-amino acids and proteins. Another interesting observation is that downregulated LPCs in the serum of cirrhotic samples found that cirrhosis may happen due to alcohol or hepatitis-B [65]. The molecular mechanisms were projected by Gonzalez and coworkers [52].

### 2.5. Hepatocellular Carcinoma and Metabolic Phenotyping 

Globally, primary liver cancer is a fatal disease that has affected more than millions of lives. Liver cancer have three main subtypes such as HCC, intrahepatic cholangiocarcinoma, and combined HCC-intrahepatic cholangiocarcinoma. Liver cancer acts as the major important cause of tumor-correlated losses. This may result from various risk factors, mainly viruses and alcohol consumption. In addition to that, NAFLD is the major risk factor that leads to the growth of HCC [66].

Metabolic stability discrimination of healthy control (HC) vs. LD and cirrhosis (CIR) vs. HCC has been performed to discover the variance between the clusters (Figure 5). Partial least squares discriminant analysis (PLS-DA) is a useful tool to extract the metabolic differences in samples. This scaling approach delivers more accuracy that inter- and intra-cluster discrepancies. The validity of the score plot has been evaluated by to cross-validation of results.

Biomarkers’ diverse metabolic features can be used to differentiate normal and anomalous molecular conditions requiring clinical therapeutic intervention via dissecting biomolecules (i.e., DNA, RNA, proteins, metabolites, etc.) [68]. Biomarker generation and disease diagnosis in HCC tumor models by metabolomic profiling is a promising technology [69]. The growth of tumor occurrence and chemical reactions are bonded with several metabolic implications.

According to evidence, metabolic modifications are a result of carcinogenesis in cancer cellular microenvironments that may need huge quantity of energy and substrate for lipid synthesis, protein transportation, and metabolic pathway transportation [70]. In HCC, it has been reported that glycolysis in tumor cells is a foundation of substrata for the pentose phosphate pathway for nucleotide synthesis, instead of delivering energy (i.e., adenosine triphosphate, ATP).

The study found that metabolic fluctuations of hypoxia-inducible factors may convert to oxygen poor situations, even in HCC tumor models. Chemical metabolomics profiling found that 4-hydroxyproline looked to be a regulating target in low oxygen survival of wild type cells, while fructose was a modifiable target in HIF (hypoxia inducible factor)- deficient cells [71]. Figure 6A shows the top 67 metabolites from various types of liver diseases and control samples. The most significant metabolites are marked in the red box.

Discovering new biomarker detection in HCC has still been complex because the inhomogeneity of cancer and pathophysiological changes resulting from potential pathogenic factors (i.e., cirrhosis, chronic inflammation, and FLD). HCC is triggered by hapatocytes involving the liver parenchyma; it happens from liver cirrhosis and chronic liver injury in 80% of clinical cases [74]. Increasingly, the root of HCC is chronic liver injury that has leads to hepatocyte regeneration and results in peripheral fibrosis and abnormal structural nodules [75].

Metabolomic profiling by GC-TOF-MS has investigated metabolic discrimination in hepatocellular carcinogenesis with random forest (RF) analysis. Metabolic dysregulation in hepatic tumorigenesis is linked with energy resources metabolisms’ macromolecular synthesis and saves tumor cells from oxidative stress [76]. Metabolic evolution from carbohydrate metabolisms to amino acid and lipid metabolism while the severity of inflammation in cirrhosis rises may happen from various metabolic pathways. Those metabolic reactions and metabolites were summarized. Targeting D-mannitol and D-glucose was found to significantly alter the HCC developmental stages [77]. Serum metabolic profiles from high-risk individuals in HCC revealed new clinical biomarkers with high sensitivity and specificity using multi-omics analysis [78].

As per a previous study, branched chain amino acid (BCAA) acts as a useful biomarker whose imbalances have been associated with the start of HCC. Significant metabolic variations in glutamic acid, citric acid, lactic acid, valine, isoleucine, leucine, alpha tocopherol, cholesterol, and sorbose levels have been reported more in HCC than liver cirrhosis [79]. Examining the metabolic changes with multi-omics platforms in HCC may resolve the pathological mechanisms. Increased levels of metabolites (i.e., 12 and 15-hydroxyeiosatetraenoic acid, xanthine, glycine, serine, aspartate, sphingosine, and acylcarnitines) have been discovered in serum samples of from healthy persons, and HCV patients by GC-MS and UPLC-MS/MS analysis. Furthermore, some of metabolites related to γ-glutamyl oxidative stress indicating anomalous cellular proliferation, neutralization responses, and pathways biosynthesis as well as eicosanoid pathways have been found to occur in the metabolisms of HCC patients [80]. According to NMR- and LC-MS-based techniques, combined with RF analysis, 32 potential biomarkers have been investigated in the serum of HCC patients, liver cirrhosis patients, and healthy volunteers. HCC can be completely detected in patients with a low 20 ng/mL of AFP values.

In Figure 6B, the Venn diagram shows the most discriminating metabolites that are associated with disease evolution (NAFLD/NASH, cirrhosis, HCC) in patients. Metabolite changes such as up- and downregulations in each disease, are marked. The metabolic activity of HCC as well as their use as targeted metabolites in various samples/platforms are detailed in Table 4. Metabolic conflicts raised in ketone biosynthesis, TCA cycle, phospholipid metabolism, sphingolipid metabolism, fatty acid oxidation, amino acid catabolism and BA metabolism in HCC are associated with various cellular molecular metabolisms [81]. An abnormal regulation of lipid metabolism may forecast disease pathology progression to HCC in metabolic syndrome patients [82].

The LC-MS method acts as an excellent diagnostic capability. Metabolomics analysis in steroid hormones, epitestosterone, and allotetrahydrocortisol were altered, which imbalanced the steroid hormone metabolic network and meaningly downregulated urinary steroid hormone levels [83]. Phenylalanine, tyrosine, glutamate, and tryptophane, kynurenine, and biogenic amines were found specifically in amino acids, which may indicate HCC progression [84]. Human biopsy tissues extract examined by NMR and MS metabolomics profiling found that aspartate metabolism leads to vital and differentiable metabolic pathways of HCC [85]. Regulation of Wnt signaling transduction pathways and various disorderly metabolites, including acyl-lysophophatidylcholine connected to HCC, is important to revise the physiological process from NAFLD to HCC [86]. Tryptophane, glutamine, and 2-hydroxybutyric acid act as beneficial biomarkers to recognize the molecular mechanisms of HCC [67,88].

## 3. Microbiome-Related Metabolites and Liver Disease

The communication between the liver and gut microbiome has developed, suggesting that the liver–gut axis is related to various developments in liver metabolic changes. Gut microbial metabolites show a vital part in the liver–gut axis or the liver–microbiome axis [90]. Various types of dominant bacterial products such as enterobacteria, bacteroides, and proteobacteria are rich in biopsy-proven liver patients [91]. Metabolomics-based analysis found an upregulation of acetinobacteria abundance but downregulation of bacteroidetes abundance in NAFLD patients [92]. Bacteria exhibit multiple effects on the development of liver disorders via saccharolytic and proteolytic fermentation in non-obese subjects. Ruminococcaceae and Veillonellaceae act as the leading microbiotas in fibrosis severity and liver fibrosis in NAFLD [93].

The short chain fatty acids (SCFAs) (acetate, propionate, and butyrate) play important metabolites in maintaining the gut barrier and intestinal homeostasis and in reducing proinflammation of cytokine secretion in the liver [94]. In intestine, SCFAs mostly are anaerobically produced by gut microbiota and regulate immunomodulatory functions. Butyrate and propionate formation in the gut act as significant energy sources for gut enterocytes, which influence the gastrointestinal barrier function via tight junction stimulation and mucous production [95].

Choline has metabolized through the microbiome from trimethylamine (TMA) to trimethylamine N-oxide (TMAO) as a final product [96]. TMA is metabolized from the intestinal microbiota via catabolism of choline, phosphatidylcholine, betaine, carnitine entering the liver by the portal vein [97]. TMAO, an oxidative product of TMA, is catalyzed via flavin-containing monooxygenases (FMO) in the liver. TMAO has been proposed to provoke the growth of NAFLD, which may happen in various metabolic mechanisms, such as tissue inflammation, dropping the BA-produced enzymes, and hepatic insulin resistance [98]. TMAO disturbs the development of NAFLD and is marked as novel biomarker for early metabolic syndrome. The metabolic changes in liver osmolytes (e.g., taurine, TMA, and TMAO) indicate that the global metabolites began to adapt to oxidative stress. TMAO may change NAFLD through BA metabolic regulation and transport. Lastly, the gut-microbiota-mediated TMA, FMO, and TMAO pathway regulates the glycolipid metabolism, cholesterol homeostasis, and hepatic inflammation [99].

Gut microbiota synchronizes secondary BA synthesis (i.e., deoxycholic and lithocholic) via cholesterol 7α-hydroxylase or oxysterol 7α-hydroxylase in the liver and stored in the gall bladder, which may be helpful to pathological regulations. Gut microbiota are involved in BA synthesis and overexpression of apical sodium-dependent BA transporter and to dysregulation of BA consumption [100].

LPS have been involved in the activation of inflammation and host immunologic systems. They are Gram-negative bacteria and act as major outer membrane component in gut. The LPS promote the occurrence and development of NAFLD and require the intestinal barrier function [101]. The metabolite of tryptophan is an important base for metabolic conversion, together with indole and its associated metabolic byproducts, by both intestinal Gram-positive and Gram-negative bacteria present in the gut environment. The byproducts of indole—such as indole-3-aldehyde, indole-3-acetic acid, and indole-3-propionic acid—gained good attention in inflammatory disorders in the liver [102]. The major routes of tryptophan metabolism, microbial degradation pathways, are affected. Gut microbiota and metabolomics analysis are hopeful techniques as therapeutic options for liver disorders. Finally, alteration in the gut–liver axis and targeting the gut bacteria related to liver disease is summarized in the Figure 7. 

## 4. Conclusions

We believe that an understanding of liver diseases such as AFLD, NAFLD, NASH, fibrosis, cirrhosis, and HCC are an important milestone toward real diagnosis of human liver and gut diseases. This review specifies that metabolomics techniques in LD and gut metabolisms have widely delivered the important information for candidate biomarker metabolites and their related metabolic pathways. The metabolomics method is simple, less expensive, and non-invasive technology.

Here we addressed how metabolomics play a role in clinical and healthcare departments. More importantly, increased understanding of the recent genome and proteome (i.e., inflammatory cytokine activity genes: IL-1beta, IL-6, and IL-8; TNF-α, TNF-β_1_) after alcohol application. Many types of metabolites in the liver environment have been quantified and targeted that have accounted for variations in energy (glucose, lactate, pyruvate), antioxidant defense systems (GSH), neurotransmission (glutamate, glutamine), catalytic activity (LDH), surfactants (Pcho, GPcho), fatty acids (triglyceride, cholesterol), and amino acids (isoleucine, valine, threonine, carnitine, phenylalanine). ROS (^•^OH, O_2_^•−^, and ^1^O_2_) are grown by alcohol intake. Those free radicals may change metabolic stability in cytoplasm, mitochondria, and nuclei, and the electron transport-chain that may trigger apoptosis. In modern days, molecular and analytical techniques are access to resources that are crucial in ensuring a stable supply metabolic chain and metagenomic imaging for medicinal efficiency. Metabolomics profiling can be a key tool for clarifying complex progress in science by quantifying what is not yet known in liver disease and the gut microbiome.

A new generation of scientists and science–policy practitioners have not yet reached the era of personalized medicine for cirrhosis and HCC. However, strong progress in science has continued to maximize the effectiveness of scientific inquiry. An evidence-based approach is good in science; metagenomics and metabolomics profiling in liver disease and the gut microbiome act as an advanced process. We are confident that high-throughput screening against the liver and gut microenvironment can be considered for cell therapies. Clinical biomarkers and metabolic screening using metabolomics profiling and metagenomics are promising tools and need to be extensively considered in the future. By synthesizing knowledge from fields such as liver, gut microbiota science, chemistry, bioengineering, and molecular biology, the groundwork has been arranged for transferring therapies into the clinic.

## Figures and Tables

**Figure 1 ijms-22-01160-f001:**
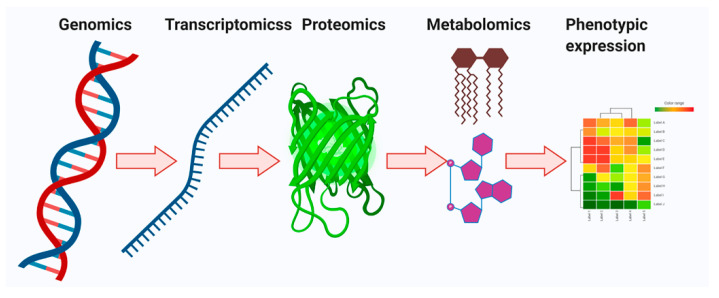
Overview of the central dogma, including fundamentals and design principles for multi-omics profiling in systems biology. The genomics, transcriptomics, proteomics, and metabolomics provide learning about DNA, mRNA, proteins, and metabolites, respectively. The multi-omics profiling in systems biology is influenced via epigenetics, age, diet, lifestyle, drugs, toxins, etc. Bilateral flow of cellular signals is detected between the genome, transcriptome, proteome, and metabolome.

**Figure 2 ijms-22-01160-f002:**
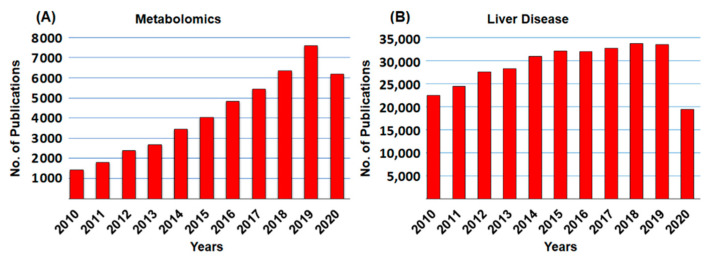
Historical development of publications in (**A**) metabolomics, and (**B**) liver diseases. Total sum of scientific publications between 2010 and 2020. The publications of metabolomics and liver diseases grew every year. All the data were accessed on PubMed on 21 September 2020.

**Figure 3 ijms-22-01160-f003:**
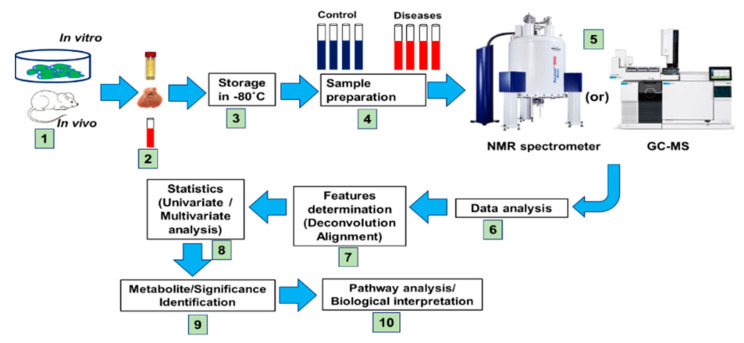
Proposed structure of symmetrical metabolomics experimental design for evaluating storage and quantitative metabolite analysis. Types of sample assessment: sample collections in tubes, data processing, statistics, and metabolic pathway analysis. The robotic high throughput metabolomics technique in liver and gut microbial/organic mixtures reveals their spectral complexity and their metabolic significance.

**Figure 4 ijms-22-01160-f004:**
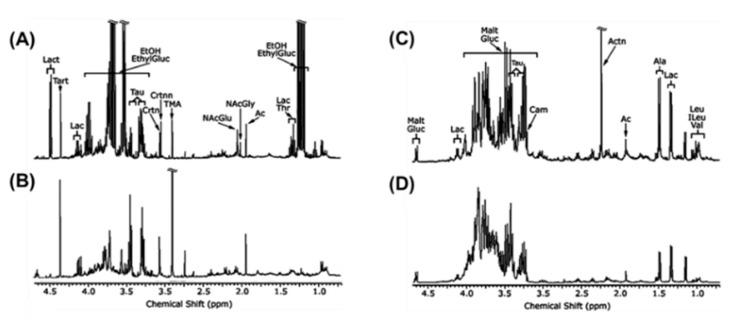
^1^H NMR spectrum of (**A**) mice urine with alcohol treatment; (**B**) mice urine with control; (**C**) mice liver with alcohol treatment; (**D**) mice liver with control. Solution state NMR experiments (400 MHz). The metabolites assignment of ^1^H NMR spectra data in alcohol-treated mice are annotated. EthylGluc, ethylglucuronide; EtOH, alcohol; Lac, lactate; Ac, acetate; NAcGly, N-acetylglycine; NAcGlu, N-acetylglutamine; TMA, trimethylamine; Crtnn, creatinine; Crtn, creatine; Tau, taurine; Tart, tartrate; Lact, lactose; Malt, maltose; Carn, carnitine; Actn, acetone; Ala, alanine; Leu, Leucine; ILeu, isoleucine; Val, valine. Adapted from reference [43] Copyright at 2008 Elsevier Inc.

**Figure 5 ijms-22-01160-f005:**
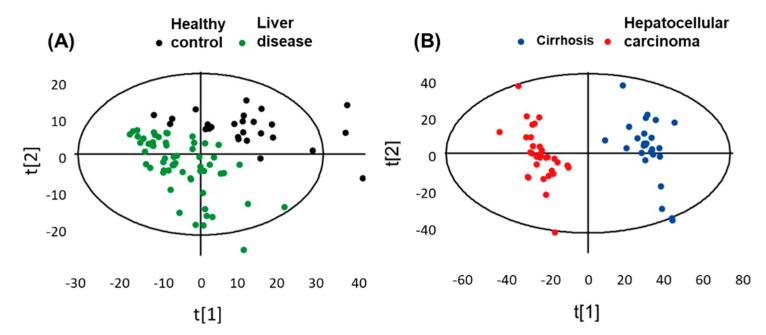
(**A**,**B**) Plot of cell metabolic discrimination by supervised PLS DA analysis. Pattern recognition of healthy control group and liver disease groups, which included cirrhosis and hepatocellular carcinoma samplesPattern recognition of healthy control group and liver disease groups, which included cirrhosis and hepatocellular carcinoma samples. Adapted with permission from [67], Copyright 2014, American Chemical Society.

**Figure 6 ijms-22-01160-f006:**
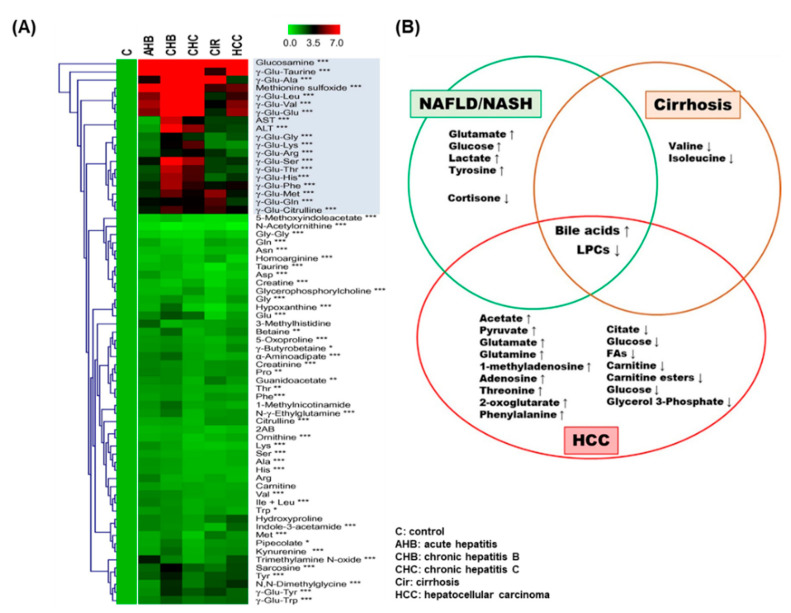
Significant metabolome in liver diseases. (**A**) 67 metabolites from various types of liver diseases. All rows and columns display the targeted metabolites regulations and liver diseases, respectively. * *p* < 0.05, ** *p* < 0.01, *** *p* < 0.0001 was calculated. Adapted from references [72], Copyright 2011, European Association for the Study of the Liver. Published by Elsevier Ireland Ltd. (**B**) The regulation of up- and downregulated metabolites in HCC, cirrhosis, and non-alcoholic fatty liver disease (NAFLD)/non-alcoholic steatohepatitis (NASH) shown in Venn diagram. Adapted with permission from [73], Copyright 2013, European Association for the Study of the Liver. Published by Elsevier B.V.

**Figure 7 ijms-22-01160-f007:**
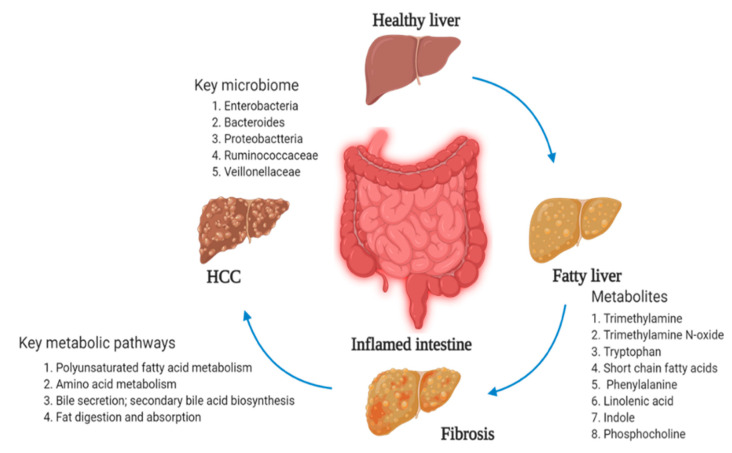
Suggested metabolic effects in gut microbiome and liver diseases.

**Table 1 ijms-22-01160-t001:** Recent analytical research technologies applied to metabolite profiling. The major omics techniques, functionalities benefits, and ability to support to metabolomics/metagenomics analysis.

Methods	Advantages	Disadvantages	Ref.
Sequencing	High throughput, massively parallel, amplifying the lowest abundant structures	Low molecular diversity: nucleic acids only	[17,18]
NMR	Nondestructive method, minimal sample preparation, quantitative analysis, tissue analysis	Lower sensitivity, low molecular diversity	[2,14,19]
MS (LC/GC)	Destructive method, capable of depicting volatile compounds, not fully quantitative, target analysis	Partial molecular diversity, less reproducible	[12,13]
CE-MS	Relatively lower cost than other methods	-	[20,21]
HPLC-MS	Extensive molecular diversity, robust	Low sensitivity	[22,23]
Raman micro spectroscopy	3D evidence, high throughput, structural information, nondestructive enabling	Low sensitivity than MS and NMR	[24,25]
Immunochemistry	Low throughput, high specificity	Targeted analysis	[26,27]

NMR, nuclear magnetic resonance; MS, mass spectrometry; CE, capillary electrophoresis; GC, gas chromatography; HPLC: high pressure liquid chromatography; LC, liquid chromatography.

**Table 2 ijms-22-01160-t002:** Metabolome, organic compounds, and gases in liver metabolisms and microbial fermentation.

Metabolome Generated	Functional Roles	Compounds
Oligomers (disaccharides, oligosaccharides), organic acids (succinate, lactate), SCFAs (acetate, propionate, butyrate, valerate), BCFAs (iso-butyrate, iso-valerate)	Sugars, starches, and fibers	Carbohydrates
SCFAs, BCFAs, biogenic amines, amino acids, phenols, p-Cresols, indoles	Structure, function, and regulation	Amino acids/proteins
Gases (CO_2_, H_2_S, NH_4_ and CH_4_), methanol, ethanol	Digestion	Gases
Conjugated fatty acids, acylglycerols, sphingomyelin, cholesterol, phosphatidylcholines, triglyceride, phosphoethanolamines	Building blocks, structure, function of living cells	Lipids/fats
Cholate, hyocholate, deoxycholate, taurocholate, chenodeoxycholate, α-muricholate, β-muricholate, ω-muricholate,	Hormonal actions, metabolic functions	Bile acids
Biotin, folate, thiamine, riboflavin, pyridoxine, vitamin K, vitamin B12	Organic molecule, micronutrient	Vitamins
Pyrocatechol, hydroxyphenyl-propionic acid, enterodiol etc.	Micronutrients, plant-based foods	Polyphenols
N-acetyltryptophan, N-acetyl cysteine, N-acetyl glucosamine	Antioxidant effects, reduce free radicals	N-acetyl compounds
Putrescine, cadaverine, spermidine	Cell proliferation, growing tissue	Polyamines

SCFAs, short-chain fatty acids; BCFAs, branched-chain fatty acids; CO_2_, carbon dioxide; CH_4_, methane; H_2_S, hydrogen sulfide; NH_4_, ammonium.

**Table 3 ijms-22-01160-t003:** Metabolome and metabolomics profiling with hepatic fibrosis.

Platform	Models	Analysis	Metabolome	Related Pathways	Ref.
UHPLC-MS GC-MS	Plasma	Box plots, Random forest importance plot	Aspartate ↑, glutamate ↑, Phenylalanine ↑, tyrosine, 3- (4-hydroxyphenyl)- lactate, kynurenine, isoleucine ↑, leucine ↑, valine ↑, ornithine ↑,	D-ornithine metabolism; Amino acid metabolism	[49]
HPLC LC-MS	Plasma	Heat map	Linolenic acid ↓, palmitoleic acid ↑, oleic acid ↑	Fat digestion and absorption	[50]
UPLC-MS	Serum	PCA	Cholic ↑, deoxycholic ↑, arachidonic acid↓, glutamic acid↓	Glycerophospholipid metabolism; choline metabolism	[51]
1H-NMR (1D)	Liver	PCA, Loading plots	Lactate ↑, choline ↑, proline ↑, Glutamine ↑, glutamate ↑, TMA ↓, glycogen ↓, inosine ↓, fumarate ↓	Glutamatergic synapse; amino acid metabolism	[53]
GC-MS	Urine	PCA, PLS-DA	Propionate ↓, benzoate ↓, leucine ↓, octanoate ↓, phenol ↓, glycine ↓, indole↓, oleic acid ↓, lysine ↓	Fatty acid metabolism; lysine degradation; lysine biosynthesis	[56]
UPLC/ESI-Q-TOF-MS	Urine	PCA	Glycocholate ↑, 2-hydoxybutanoic acid ↓	Bile secretion; secondary bile acid biosynthesis	[57]
^1^H-HR-MAS-NMR (1D, 2D)	Liver	PLS-DA, Loading plots	Phosphocholine ↑, Phosphoethanolamine ↑, glutamate ↑,	Glycerophospholipid metabolism;	[58]
HPLC-LTQ-MS	Serum	OPLS-DA, column plot	glycolchenodeoxycholic acid ↑, lysophosphatidylcholine ↑	-	[59]
^1^H-NMR (1D)	Serum	PCA, PLS-DA	Acetate ↑, pyruvate ↑, Glutamine ↑, taurine ↑, 2-oxoglutarate, glycerol ↑, tyrosine ↑, phenylalanine ↑, 1-methylhistidine ↑	Phenylalanine metabolism, D-Glutamine and D-glutamate metabolism; citrate cycle (TCA cycle); tyrosine metabolism	[60]
^1^H-NMR (1D)	Serum	PCA, OPLS-DA, loading plots	Isoleucine ↓, valine ↓, phenylalanine ↑, formate ↑, acetate ↑, lysine ↓	Valine, leucine, and isoleucine biosynthesis	[61]
GCxGC-TOF-MS	Serum	R^2^ values	D-alanine ↓, D-proline ↓	Arginine and proline metabolism; amino acid metabolism	[62]
^1^H-NMR (1D)	Serum	PCA, loading plots, heat map	Glucose ↓, lactate ↑, choline ↓, VLDL/LDL ↓,	Polyunsaturated fatty acid metabolism	[63]

HPLC, ultra-performance liquid chromatographic; PCA, principal component analysis; NMR, nuclear magnetic resonance; MS, mass spectrometry; OPLS-DA, orthogonal PLS-DA; GC, gas chromatography; HPLC: high pressure liquid chromatography; LC, liquid chromatography; VLDL, very-low-density lipoprotein.

**Table 4 ijms-22-01160-t004:** Metabolome and metabolomics profiling in hepatocellular carcinoma.

Platform	Sample	Analysis	Metabolites	Related Pathways	Ref.
LC-MS	Urine	PLS-DA, Heat map, ROC curve	Nucleosides, bile acids, citric acid, amino acids, cyclic adenosine monophosphate, glutamine, acylcarnitines	Purine metabolism, energy metabolism, amino acid metabolism	[67]
GC-TOF-MS	Serum	PCA, OPLS-DA, heat map	Phenylalanine, malic acid, 5-methoxytryptamine, palmitic acid, asparagine, b-glutamate	Energy metabolism, macromolecular synthesis, oxidative stress	[76]
GC-MS	Cells	Heat map, loading plots	D-mannitol, D-glucose	Lipid and amino acid metabolism	[77]
LC-MS	Serum	PLS-DA, ROC curves	Xanthine, uric acid, cholyglycine, D-leucic acid, 3-hydroxycapric acid, arachidonyl lysolecithin, dioleoyl phosphatidylcholine	Purine catabolism lipid metabolism	[78]
GC-MS	Plasma	PLS-DA, OPLS-DA	Glutamic acid, citric acid, lactic acid, valine, isoleucine, leucine, alpha tocopherol, cholesterol, sorbose	Branched-chain amino acid metabolism	[79]
GC-MS, UPLC-MS-MS	Serum		12-HETE, 15-HETE, sphingosine, xanthine, amino acids serine, glycine, aspartate, acylcarnitines	Cell regulation, amino acid biosynthesis, neutralization reaction, eicosanoid	[80]
^1^H-NMR (1D), LC-MS	Serum	PCA, random forests analysis	Formate, tyrosine, ascorbate, oxaloacetate, carnitine, phenylalanine, C16 sphinganine, lysophosphatidylcholines, phosphatidylcholines	Ketone biosynthesis, citric acid cycle, phospholipid, fatty acid oxidation, sphingolipid, amino acid/bile acid metabolism	[81]
GC-MS	Serum Liver tissues	Gene expression	Triglycerides, cholesterol, fatty acids	Lipid metabolism	[82]
LC-MS	Urine	PCA, heat map	Epitestosterone, allotetrahydrocortisol	Steroid hormonal system, steroid hormone pattern	[83]
LC-MS	Serum	Spearman correlation	Phenylalanine, tyrosine, glutamate, kynurenine, tryptophan, biogenic amines	Amino acid, biogenic amine metabolism	[84]
^13^C-NMR, LC-MS/MS	Tissues	PCA	Alanine, succinate, lactate, glycerophosphoethanolamine, inorganic phosphate, leucine, isoleucine, valine	Aspartate metabolism, tricarboxylic acid metabolism	[85]
LC-MS	Liver	Gene expression	Lysine, phenylalanine, citrulline, creatine, creatinine, inosine, glycodeoxycholic acid, alpha-ketoglutarate, multiple acyl-lyso-phosphatidylcholine	Krebs cycle, urea cycle, amino acid, purine metabolism	[86]
UHPLC-MS	Serum	PCA	Acylcarnitines, fatty acids, phosphatidyl ethanolamine	Fatty acid, b-oxidation, phosphatidylcholine, phosphatidyl ethanolamine metabolism	[87]
CE-TOF/MS	Serum	PCA, PLS-DA, correlation network	Creatine, betaine, kynurenine, pipecolic acid	Fundamental carbon metabolism, glycerolipid digestion, methylation reactions, oxidative stress	[88]
CE-TOF/MS	Serum	PLS-DA, ROC curves, V-plot	Tryptophan, glutamine, and 2-hydroxybutyric acid	Amino acid metabolism	[89]

HPLC, ultra-performance liquid chromatographic; ROC, receiver operating characteristic; PCA, principal component analysis; NMR, nuclear magnetic resonance; MS, mass spectrometry; OPLS-DA, orthogonal PLS-DA; GC, gas chromatography; HPLC: high pressure liquid chromatography; LC, liquid chromatography; CE, capillary electrophoresis; v, volcano.

## Data Availability

Data is contained within the article.

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
