# Peer review of "Recent Advances of Microbiome-Associated Metabolomics Profiling in Liver Disease: Principles, Mechanisms, and Applications"

_ijms, 2021, doi:10.3390/ijms22031160_

Round 1
Reviewer 1 Report
The submitted contribution have announced a review focused on the metabolomic profiling in liver and gut microbiota, particularly on principles, mechanisms and applications.
Nevertheless, reading of the manuscript does not bring the expected benefits to the reader. The work suffers from the absence of a clear concept and a logical knowledge management from the reviewed last 10 year period. The text is full of general phrases not useful to the assumed auditorium and some important issues have not been addressed. The English language is also not satisfactory in some partsd and should be improved.
The general comments
1. The conceptual issues
Metabolomics profiling in liver, liver tissues, body fluids or that of liver diseases ?
...in gut microbiota. I did not find relevant knowledge about this topic in the text.
2. Introduction - about metabolomics
L.43 Metabolomics techniques are much more advantageous than - omics techniques .... a phrase without a content. Why they are more advantageous ?
Provided that metabolomics of liver diseases would be the main goal, then a brief characterization of liver tissue and the studied diseases should be briefly described aiming to attract the reader to the review subject. Furthermore, the successful metabolomics strategies should be described, starting from the biological material collection, extraction, instrumental analysis to the data processing. The pitfalls and personal experience with the applied metabolomics approaches will be appreciated.
3. Defect metabolomics
2.1. Overview
A rather confusing part full of the phrases without a scientific content.
Table 1
e.g.
MS (coupled with LC....) highly sensitive ???
compare to HPLC-MS ..... comparatively high sample required ??
low sensitivity ???
Immunochemistry in fact, lower specificity via cross reactions....
CE-MS vs. LC-MS is really low cost ???
Liver disease
The text contains plenty of formal published paper descriptions. Instead,
the disease, the affected metabolite pathway and key disturbed metabolites in central and lipid metabolism could be highlighted.
2.2. AFLD and phenomic
Why is the general Fig.3 in this section ??
2.3.NAFLD, NASH
Can we deduce from the literature the most relevant metabolites associated with the particular liver diseases ?
2.4. Liver fibrosis, cirrhossis
Table 3, rather than the metabolite list, synthesis of the recent knowledge resolving the upregulated and downregulated events associated with the liver diseases should be assessed.
2.5. Carcinomas
Similarly, rather than reproduction of the reported data, the reader would expect description of some general outcomes obtained from the literature review. Classification according to particular pathways would be profitable, for instance from the energy metabolism, over BCAA to lipid, steroid perturbations.
Reproduction of the earlier data is of less use. Extraction of the key observations and their synthesis would be highly useful and beneficial for the Reader.
Conclusions.
The review mission aims at that, what is impossible to accomplish in a single work; (i) to describe metabolomics techniques, the field, where the authors are evidently not fully at home, (2) the diseases and the studied pathways and related metabolites and (3) the more or less formal description of various applications.
Often less is sometimes more. I think that this work is this case.
Author Response
MDPI_ IJMS_1067964
Comments from the editors and reviewers:
First of all, we would like to thank the Reviewer1 for his/her comments, which helped us to improve this manuscript.
- Comment 1: The submitted contribution have announced a review focused on the metabolomic profiling in liver and gut microbiota, particularly on principles, mechanisms and applications. Nevertheless, reading of the manuscript does not bring the expected benefits to the reader. The work suffers from the absence of a clear concept and a logical knowledge management from the reviewed last 10 year period. The text is full of general phrases not useful to the assumed auditorium and some important issues have not been addressed. The English language is also not satisfactory in some partsd and should be improved.
Answers: We are grateful for the reviewer’s valuable comments. As per your concerns, we added the following details in manuscript and revised whole manuscript as reviewer pointed (we noted that in track change marked version). In addition, we changed and added new points in liver disease and microbiome associated metabolites (3. Microbiome related metabolites and liver disease).
In this review, we carefully elaborate the metabolomics profiling in various liver diseases such as fatty liver diseases (FLD), non-alcoholic steatohepatitis (NASH), fibrosis hepatic steatosis, and cirrhosis, and quantified metabolites and related pathways. Finally, very important candidate gene/protein and metabolites for liver disease and related gut microbiota potential were discussed in this review points. From last 10 years, more cited publications had been considered and cited which will be the effective liver and gut therapy in human liver cancer cells, NASH, hepatic steatosis, and cirrhosis by metabolomics analysis. Per every year, publications of metabolomics analysis in liver disease are delivered more than thousands of publications. Among them, extracting the knowledge and understanding the concept is truly hard. However, we cited more relevant literatures in this review articles. As an author, we are design the review manuscript how scientific methods such as NMR, MS, Raman, and Immunochemistry are working in metabolomics analysis in liver diseases and gut microbiota role.
In addition, we done official language edition before summation to the journal. Here, editing certificate proof is attached for your clarifications. Thanks for your comments.
- Comment 2: The general comments 1. The conceptual issues. Metabolomics profiling in liver, liver tissues, body fluids or that of liver diseases?...in gut microbiota. I did not find relevant knowledge about this topic in the text.
Answers: Thank you for your valuable comments. As per your concerns, abstract is carefully revised. This manuscript reviewed in metabolites role in liver diseases and metabolites involved in gut environment. Table 2 metabolites involved in gut microbial fermentation and liver metabolisms. In addition, we added some explanation about liver disease and metabolites.
“The gut microbial community consists eco-system with the functions in host protection against pathogens, immune modulation, metabolic pathways, and enterohepatic circulation. As liver is directly connected with the gut through the portal vein, gut-derived toxic factors including bacteria, damaged metabolites (damage-associated molecular patterns), or bacterial products (pathogen-associated molecular patterns) are metabolized in the liver [15]. Some gut microbiota produce ammonia, ethanol, and acetaldehyde, which are mostly metabolized in the liver and are associated with Kupffer cell activation and the inflammatory cytokine pathway [16]. Interests in metabolites is increasing as gut microbiome related-metabolites are key pathophysiologic factor in the liver disease (LD) progression.
Gut bacteria genera might involve in the fermentation biology of polysaccharides, energy collecting, bile acid (BA) synthesis, and choline metabolism in liver cells. Bacterial changes such as Enterobacteria, Bacteroides, Proteobactteria, Faecalibacterium, Ruminococcus, Lactobacillus, and Bifidobacterium has been mirrored to liver metabolites and their metabolic reaction network. Liver cirrhosis relates to bile secretion disorders and metabolic syndrome. Gut microbiota has involved to ameliorate the liver diseases. In this review, the analysis technology, specific metabolites, and application fields of metabolomics in liver disease were described.”
- Comment 3: 2. Introduction - about metabolomics L.43 Metabolomics techniques are much more advantageous than - omics techniques .... a phrase without a content. Why they are more advantageous?
Answers: We apologize for causing confusions. We erased sentence and explained Metabolomics techniques. Metabolomics techniques, maturing at a time, play the comprehensive analysis of small-molecule metabolites under a given set of conditions. Metabolites serve as direct signatures of metabolic reaction and biochemical activity. To study the biological mixtures of attention spectroscopic (NMR), spectrometric (MS) and separation techniques (GC, LC, CE) are applied. These analytical techniques play a greater role in the clinical metabolomics profiling studies. Several small molecules, organic compounds from various cellular microenvironments have been quantified and profiled the metabolites to understand the metabolic reaction network.
- Comment 4: Provided that metabolomics of liver diseases would be the main goal, then a brief characterization of liver tissue and the studied diseases should be briefly described aiming to attract the reader to the review subject. Furthermore, the successful metabolomics strategies should be described, starting from the biological material collection, extraction, instrumental analysis to the data processing. The pitfalls and personal experience with the applied metabolomics approaches will be appreciated.
Answers: Thank you very much for your comments. As per your concerns, our goal and purpose of this review is that understanding the various categories of liver disease and with metabolomics profiling techniques. The brief explanation and illustration of metabolomics strategies, sample preparation, data analysis, statistics, and metabolite identifications are summarized in figure. 3. It will briefly be describing the fundamental structure of metabolomics experimental approaches from step 1 to step 10.
- Comment 5: 3. Defect metabolomics 2.1. Overview A rather confusing part full of the phrases without a scientific content.
Answers: As per your comments, we are revised the manuscript that prepared without damaging the structure, shape, and quality of science in this review works. Thanks.
- Comment 6: Table 1 e.g. MS (coupled with LC....) highly sensitive ??? compare to HPLC-MS ..... comparatively high sample required ?? low sensitivity ??? Immunochemistry in fact, lower specificity via cross reactions....CE-MS vs. LC-MS is really low cost ???
Answers: We agree with the reviewer’s comment and apologize for causing confusion. In Table 1, high-throughput analytical methods including LC/GC-MS and NMR applications have also been frequently compared to investigate and quantify the metabolome of given samples. The comparison study shows the advantages of analytical techniques. NMR used to quantitatively analyze mixtures and molecular structure. For your explanation, the functional role of high-resolution mass spectrometry (MS; coupled with LC and GC) techniques is drafted in Table-1. Price is comparatively low with another machine. The sensitivity of NMR that intrinsically low but can be improved with multiple scans (times), field strength, cryo-cooled, microprobes, and hyperpolarization methods. The sensitivity of MS that has highly sensitive with detection limit in the nanomolar (nM) range. Here, the MS, the term sensitivity can have several meanings that are often used interchangeably. Sensitivity may be defined as the change in signal per unit change in concentration of an analyte (such as the slope of the calibration curve). More commonly, it is used to reference the magnitude of the signal produced by the analyte in the MS detector. MS sensitivity is often used to compare detectors.
- Comment 7: Liver disease The text contains plenty of formal published paper descriptions. Instead, the disease, the affected metabolite pathway and key disturbed metabolites in central and lipid metabolism could be highlighted.
Answers: Much thanks for your comments. The manuscript is written as per our understanding literature review in this domain. We have been revised with suitable novel publications that are utilizing as cited reference. Understanding the liver disease and gut microbiota in this manuscript and worked as carefully in throughout the manuscript from others. As per your suggestions, revised the central and lipid metabolism in this manuscript. Thanks.
- Comment 8: 2.2. AFLD and phenomic Why is the general Fig.3 in this section ??
Answers: As reviewer pointed out, we changed location of Fig 3. Overview of metabolomics workflow shown in Figure 3. The important fundamental steps and evaluating the metabolites are here proposed which may highly be useful to metabolomics readers through this manuscript. An automated high-throughput metabolomics platform is used to elucidate molar concentrations of organic compounds, reveal their spectral data, and generate pathway analyses.
- Comment 9: 2.3.NAFLD, NASH Can we deduce from the literature the most relevant metabolites associated with the particular liver diseases ?
Answers: Thank you for your valuable comments. The relevant publications in NAFLD, and NASH are cited. It provides the details about metabolome, metabolic reaction network, and associated metabolic reaction network.
- Comment 10: 2.4. Liver fibrosis, cirrhosis Table 3, rather than the metabolite list, synthesis of the recent knowledge resolving the upregulated and downregulated events associated with the liver diseases should be assessed.
Answers: Table -3 is revised as per suggestion. Metabolites regulations are marked. Thanks.
- Comment 11: 2.5. Carcinomas Similarly, rather than reproduction of the reported data, the reader would expect description of some general outcomes obtained from the literature review. Classification according to particular pathways would be profitable, for instance from the energy metabolism, over BCAA to lipid, steroid perturbations. Reproduction of the earlier data is of less use. Extraction of the key observations and their synthesis would be highly useful and beneficial for the Reader.
Answers: Much thank for your comments. As per your suggestions, in this review, we collected metabolites and related metabolic pathways from suitable publications in the domain of liver disease and gut microbiota. Here, we discussed the metabolome and associated metabolic reaction based on metabolomics of liver disease such as AFLD, NAFLD, NASH, liver fibrosis, cirrhosis, and hepatocellular carcinoma (HCC). Every disease related significant metabolic profiling and pattern recognition analysis are delivered in this review which would be great benefit to general readers. We collected the figures with citation from other publications and discussed in our ways. We described our points in every section with suitable outcomes of data. Thanks.
- Comment 12: Conclusions. The review mission aims at that, what is impossible to accomplish in a single work; (i) to describe metabolomics techniques, the field, where the authors are evidently not fully at home, (2) the diseases and the studied pathways and related metabolites and (3) the more or less formal description of various applications. Often less is sometimes more. I think that this work is this case.
Answers: Thank you for your valuable comments. The goal of our work is described the basic applications of (1) metabolomics analysis, and pattern recognition analysis, (2) metabolic reaction pathways has explained in every disease-based analysis, (3) listed quantification of metabolites from different platforms (NMR, GC-MS, LC-MS, HPLC, etc.,). Here, many types of metabolites in liver environment have quantified and targeted that has accounted. In this manuscript covers the of energy (glucose, lactate, pyruvate), antioxidant defense systems (GSH), neurotransmission (glutamate, glutamine), catalytic activity (LDH), surfactant (Pcho, GPcho), fatty acid (triglyceride, cholesterol), amino acids (isoleucine, valine, threonine, carnitine, phenylalanine).

Reviewer 2 Report
The review article by Raja G et al., summarized the recent advancement in the technologies used to study liver disease phenotype. It presents an excellent overview on the topic. The authors carefully discussed the interpretation of data from various technologies in both alcoholic and non-alcoholic fatty liver diseases. The topic is relevant and is worth a review of literature. The review is of good quality, encompassing a wide range of good evidence. However, it tends to be a little too long, in particular for the presence of broad explanations of general concepts. This aspect could reduce the appeal of the paper in an average audience of clinicians.
Author Response
MDPI_ IJMS_1067964
Comments from the editors and reviewers:
First of all, we would like to thank the Reviewer2 for his/her comments, which helped us to improve this manuscript.
Reviewer 2
Comments and Suggestions for Authors
The review article by Raja G et al., summarized the recent advancement in the technologies used to study liver disease phenotype. It presents an excellent overview on the topic. The authors carefully discussed the interpretation of data from various technologies in both alcoholic and non-alcoholic fatty liver diseases. The topic is relevant and is worth a review of literature. The review is of good quality, encompassing a wide range of good evidence. However, it tends to be a little too long, in particular for the presence of broad explanations of general concepts. This aspect could reduce the appeal of the paper in an average audience of clinicians.
Answer to Reviewer-2: Much thanks for these wonderful scientific points of revisions and appreciation of our review works. As per your scientific comments, we are revised the manuscript that prepared without damaging the structure, shape, and quality of science in this review works.
Finally, we all are revised and approved the technical comments. Much thanks to all the reviewers for their scientific commentaries.
